

# Triple stable isotope analysis to estimate the diet of the Velvet Scoter (*Melanitta fusca*) in the Baltic Sea

Rasa Morkūnė[1], Jūratė Lesutienė[1], Julius Morkūnas[1] and Rūta Barisevičiūtė[2]

[1] Marine Research Institute, Klaipeda University, Klaipeda, Lithuania
[2] Center for Physical Sciences and Technology, Vilnius, Lithuania

## ABSTRACT

This study quantifies contributions of different food sources in the winter diet of the Velvet Scoter (*Melanitta fusca*) in coastal waters of the Lithuanian Baltic Sea using non-lethal avian sampling. We highlight the application of stable sulphur isotope ratios as complementary to stable carbon and nitrogen isotope analysis in order to discriminate sandy bottom macrozoobenthos organisms as potential food sources for the Velvet Scoter. Selection of the most relevant trophic enrichment factors and Monte Carlo simulations in order to choose the best fitted model were undertaken. The stable isotope mixing model revealed the main contributions of a group of bivalves, *Mya arenaria* and *Cerastoderma glaucum*, to be 46–54%, and while the crustacean, *Saduria entomon*, comprised 26–35% of the diet.

## INTRODUCTION

Many studies have revealed relationships between the distribution of wintering marine ducks and macrozoobenthos communities (*Kube, 1996*; *Loring et al., 2013*; *Žydelis et al., 2009*). Anthropogenic activities such as commercial harvesting of benthic organisms, trawling, development of wind parks, introduction of new species, eutrophication, and climate change might have negative consequences on the composition and productivity of benthic communities. Alterations in the availability of feeding resources or the extent of feeding habitat degradation have been identified as important issues contributing substantially to the decline in the number of wintering ducks in the Baltic Sea (*Skov et al., 2011*). However, they have not been directly reported for the Velvet Scoter (*Melanitta fusca*), although regular observations of the winter diet composition and foraging grounds of this species might be important for an analysis of declines and conservation management.

The Velvet Scoter is considered a vulnerable species over its entire distribution (*BirdLife International, 2016*). In the Baltic Sea, the total number of its wintering population was reported as having decreased by 60% over the last two decades (*Skov et al., 2011*). Mid-winter surveys in the Lithuanian coastal zone of the south-eastern Baltic Sea showed an 80% decline (from 40,000 to 8,000) in wintering scoters (*Švažas, 2001*; *Šniaukšta, 2012*; *Šniaukšta, 2014*; *Šniaukšta, 2015*; *Šniaukšta, 2016*). These midwinter estimates of scoters

Corresponding author
Rasa Morkūnė,
rasa.morkune@apc.ku.lt

did not include concentrations known to be offshore at depths <35 m (*Daunys et al., 2015*). Nevertheless, inadequate studies of trophic ecology, limit understanding of the factors controlling changes in the distribution and number of wintering Velvet Scoters.

Outside the breeding period, Velvet Scoters mainly feed upon marine bivalves that live on the surface or within the upper sandy substrates <20 m deep. Crustaceans, including isopods and amphipods, annelids, echinoderms and fish had been also found in the oesophagus contents (*Žydelis, 2002*; *Fox, 2003*). Since a single species can often dominate the scoter's diet, this food must be of sufficient local abundance to fulfil the nutritional needs of ducks. It is assumed that most scoters feed in shallow areas, where the highest density of suitable prey biomass occurs. Moreover, flights by scoter flocks among different coastal areas likely help to find the best feeding habitats. Research on habitat use and foraging ecology through direct observations is difficult to conduct in a marine environment, so that most studies of seaduck foraging ecology and diet have been based mostly on analysis of gut contents from bycaught specimens (*Duffy & Jackson, 1986*; *Fox, 2003*; *Barrett et al., 2007*).

Declining wintering populations of scoters have led to fewer bycaught birds available for dietary studies. Moreover, insufficient fishery regulations and a protection status targeted towards a zero bycatch mortality led to an unwillingness of fisherman to deliver specimens they have caught for scientific studies. This has resulted in a search for alternative non-lethal methods to investigate the feeding habits of marine birds. Stable isotope analysis (SIA) of blood samples from living birds provides opportunities for non-lethal dietary studies, which is important for the protection of threatened species and ethical reasons (e.g., *Jardine et al., 2003*; *Cherel et al., 2008*; *Morkūnė et al., 2016*). The stable isotope (SI) approach has been widely applied to estimate energy flows and food web interactions. However, this method has been particularly powerful when isotopic patterns ('isoscapes') in a study ecosystem are known and the appropriate food sources differ isotopically among each other (*Phillips, Newsome & Gregg, 2005*). In the Baltic Sea, riverine discharge and nitrogen-fixing cyanobacteria blooms complicate isotopic differentiation between stable carbon ($^{13}C/^{12}C$, $\delta^{13}C$) and nitrogen ($^{15}N/^{14}N$, $\delta^{15}N$) isotopes because of highly variable SI values in the primary organic matter sources (*Rolff & Elmgren, 2000*; *Antonio et al., 2012*; *Lesutiene et al., 2014*). However, our previous study on the inclusion of sulphur ($^{34}S/^{32}S$, $\delta^{34}S$) isotopes in analysis of Baltic Sea food webs (*Morkūne et al., 2016*) revealed the possibility to distinguish food sources that were either derived from benthic production influenced by sulphur reduction, or from pelagic well-oxygenated water layers (*Connolly et al., 2004*; *Croisetiere et al., 2009*; *Fry & Chumchal, 2011*).

This study aims to quantify the contributions of different food sources in the winter diet of the Velvet Scoter based on triple SIA in blood samples in the Baltic Sea. It highlights the application of $\delta^{34}S$ as complementary to $\delta^{13}C$ and $\delta^{15}N$ ratios to discriminate sandy bottom macrozoobenthos organisms as potential food sources for the Velvet Scoter. Gut content analyses from bycaught Velvet Scoters was used to verify and complement SI mixing model results.
## METHODS

### Study site

The study site is located in the Lithuanian coastal zone of the south-eastern Baltic Sea. It is an open coastal area with dominant sandy benthic habitats which serve as important wintering grounds for Western Palearctic concentrations of the Velvet Scoter. Due to permanent sand transfer, wave and current actions, as well as the absence of macrophytes and boulders, benthic species biomass in the shallow mobile sand habitat <6 m depth is low and dominated by burrowing infaunal (polychaetes, bivalves *Macoma balthica*) and actively swimming nectobenthic common shrimps (*Crangon crangon*). The deeper (up to 30 m depth) benthic community is mostly represented by *M. balthica*, *Mya arenaria*, *Cerastoderma glaucum*, polychaetes, and nectobenthic isopods *(Saduria entomon)* (*Olenin & Daunys, 2004*).

### Collection of ducks from fishery bycatch and gut content analysis

Diet composition was estimated for 71 Velvet Scoters. These birds drowned in gillnets during regular fishery activities throughout March and November of 2012 and from November 2015 to April 2016 at depths ranging from 2 to 22 m above the sandy benthic habitat (Fig. 1). Carcasses were supplied voluntarily by coastal commercial fishermen. In a laboratory, esophagi and gizzards contents were sorted by animal prey item. Most collected birds contained some pebbles, which were excluded from further calculations. The diet composition was assessed according to the total wet weight (g) of prey and the proportion of the total wet weight (%), including mollusc shells. The ash-free dry weight (AFDW) of the prey in grams and percent represented a measurement of the weight of organic material and was calculated according to *Rumohr, Brey & Ankar (1987)* and *Timberg et al. (2011)*. The frequency of the occurrence of various prey items found in gut contents was expressed as percent of the total number of ducks used for the diet analysis.

### Sample collection for stable isotope analysis and measurements

Wintering velvet scoters were captured using the night lighting technique (*Whitworth et al., 1997*) from November 2012 to February 2013 over waters 5–15 m in depth in the Lithuanian coastal zone (Fig. 1). Permits to capture, use and release birds were obtained from the Environmental Protection Agency of Lithuania (No 7, 2012, and No 1, 2013). Blood (0.5–1 ml) was obtained from the medial metatarsal vein of live birds (*Arora, 2010*). The blood samples were stored frozen at −20 °C in cryogenic vials. Whole blood samples were freeze-dried for 48 h, weighed, and placed in tin capsules (0.5–0.7 mg for carbon and nitrogen, 1.7–2.0 mg for sulphur) for SIA.

Macrozoobenthic organisms were collected for SIA in two foraging areas important for velvet scoters in December 2012 (Fig. 1). A Van Veen sampler was used to collect macrozoobenthos (bivalves, polychaetes) in the coastal sandy bottom area at a depth range from 10 to 15 m. However, as crustaceans *S. entomon* were not found in the samplers, they were collected from a scientific bottom trawl on the sandy Klaipeda-Ventspils Plateau at a depth of 35 m in the northern part of the study site. Information about the distribution and biomass of *S. entomon* is not extensive for the Lithuanian coastal zone because the species
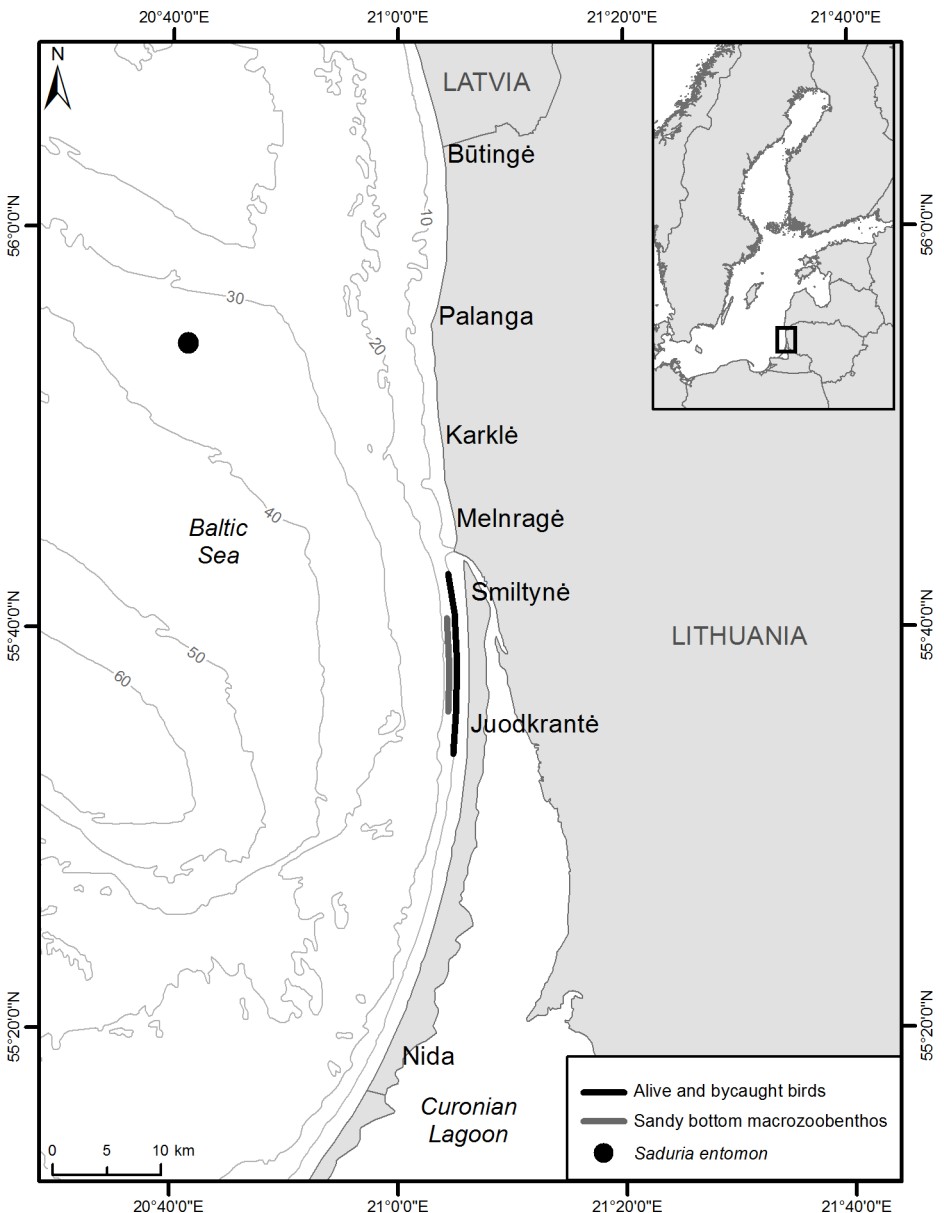

**Figure 1** **Sampling locations of alive velvet scoters and prey items for stable isotope analysis and by-caught velvet scoters for gut content analysis.**

prefers deeper habitats in the Baltic Sea. However, it is known that after disruption of the thermocline during the second part of winter, *S. entomon* migrate to near-shore coastal areas (*Bacevičius, 2013*) and become available for coastal predators such as benthivorous ducks and fish (based on preliminary stomach analysis; *Žydelis, 2002*; *Šiaulys et al., 2012*), including the area where birds were caught for this study. Moreover, the area where *S. entomon* were sampled, has been designated as an important marine area for marine birds, particularly due to their stable numerous concentrations during winter time (*Daunys et al., 2015*). As we assume that *S. entomon* must be available prey on the coastal zone at least

at the second part of winter, and that Velvet Scoters could move between main coastal areas and deeper sandy Klaipeda-Ventspils Plateau, the *S. entomon* sampling site were representative for this study.

Entire polychaetes, muscle tissue of crustaceans, and soft tissues of bivalves were taken for SIA. The sampled material was dried at 60 °C for 48 h and then was stored frozen until analysis. Unfrozen samples were ground into a fine powder in an agate mortar, weighed and placed into tin capsules (0.5–0.7 mg for $\delta^{13}C$ and $\delta^{15}N$ analysis and 1.1–2.3 mg for $\delta^{34}S$ analysis).

Isotope-ratio analysis involved precise measurement by mass spectrometry of the less abundant heavy isotope relative to the more abundant light isotope ($^{13}C/^{12}C$, $^{15}N/^{14}N$, and $^{34}S/^{32}S$) of the carbon dioxide ($CO_2$), nitrogen gas ($N_2$), and sulphur dioxide gas ($SO_2$) generated from the combustion of the sample material. The $\delta^{13}C$ and $\delta^{15}N$ ratios in the samples were determined using a Thermo Scientific Delta V Advantage mass spectrometer coupled to a Flash EA 1112 elemental analyser at the State Research Institute Center for Physical Sciences and Technology, Lithuania. The $\delta^{34}S$ values were determined using a Ser-Con elemental analyser and custom cryofocusing system interfaced to a SerCon 20-22 IRMS (Sercon Ltd., Cheshire, UK) at the Stable Isotope Facility, University of California, USA.

The results of the isotopic ratios were compared to conventional standards, i.e., Vienna Peedee Belemnite (VPDB), for carbon, atmospheric $N_2$ for nitrogen, and Vienna Canyon Diablo troilite (VCDT) for sulphur, defined as $\delta$ values: $\delta X = [(R_{sample}/R_{standard}) - 1] \times 10^3$ (‰), where $X = {}^{13}C$, $^{15}N$ or $^{34}S$, and $R = {}^{13}C/{}^{12}C$, $^{15}N/^{14}N$ or $^{34}S/^{32}S$. For calibration of the $CO_2$ and $N_2$ reference gases, the international standards from the International Atomic Energy Agency (Vienna) were used: IAEA-600 (Caffeine, $\delta^{13}C = -27.771 \pm 0.043‰$ VPDB) and NBS-22 (Oil $\delta^{13}C = -30.031 \pm 0.043‰$ VPDB) were used for $^{13}C$ and IAEA-600 (Caffeine, $\delta^{15}N = 1 \pm 0.2‰$ airN2) for $^{15}N$. Repeated analyses of the homogeneous material yielded standard deviations of less than 0.08‰ for carbon and 0.2‰ for nitrogen. For calibration of the $SO_2$ reference gases, three laboratory standards were calibrated directly against IAEA-S-1 (Silver Sulphide, $\delta^{34}S = -0.30‰$ VCDT), IAEA-S-2 (Silver Sulphide, $\delta^{34}S = 22.7 \pm 0.2‰$ VCDT), and IAEA-S-3 (Silver Sulphide, $\delta^{34}S = -32.3 \pm 0.2‰$ VCDT) were used. Repeated analysis of the three laboratory standards yielded standard deviations of less than 0.3‰. The long-term reproducibility of $\delta^{34}S$ measurements is $\pm$ 0.4‰.

Lipid removal in the benthic samples was not performed in order to keep the $\delta^{15}N$ values unaffected by treatment (*Post et al., 2007*). The C:N ratios the in majority of the benthos samples were higher than the recommended limit for aquatic organisms (C:N>3.5), at which a lipid correction should be performed (Table 1). Therefore, we corrected their $\delta^{13}C$ values using an arithmetic lipid normalization equation proposed by *Post et al. (2007)*: $\delta^{13}C = \delta^{13}C_{untreated} - 3.32 + 0.99 \times$ C:N. Lipid correction for bird blood was not applied (*Cherel, Hobson & Hassani, 2005*).

## Analysis of stable isotope ratios
The SPSS statistical software (SPSS/7.0; SPSS, Chicago, IL, USA) and R software (*R Core Team, 2013*) were used for the calculations and presentations of the results.

**Table 1  Macrozoobenthos organisms as the food sources for the mixing models of the Velvet Scoters.**

| Sources | Sample size for $\delta^{13}$C&$\delta^{15}$N/$\delta^{34}$S | C:N | $\delta^{13}$C$_{untreated}$, ‰ | $\delta^{13}$C, ‰ | $\delta^{15}$N, ‰ | $\delta^{34}$S, ‰ |
|---|---|---|---|---|---|---|
| *Saduria entomon* | 6/6 | $6.1 \pm 0.4$ | $-21.5 \pm 0.3$ | $-18.8 \pm 0.3$ | $13.1 \pm 0.3$ | $15.1 \pm 0.9$ |
| *Crangon crangon* | 6/6 | $3.4 \pm 0.0$ | $-20.1 \pm 0.1$ | $-20.1 \pm 0.1$ | $12.4 \pm 0.2$ | $13.5 \pm 0.9$ |
| *Macoma balthica* | 6/6 | $4.8 \pm 0.1$ | $-22.8 \pm 0.1$ | $-21.5 \pm 0.2$ | $7.8 \pm 0.2$ | $10.7 \pm 0.2$ |
| *Mya arenaria* | 9/5 | $4.1 \pm 0.1$ | $-22.6 \pm 0.3$ | $-21.9 \pm 0.2$ | $6.6 \pm 0.2$ | $16.6 \pm 0.6$ |
| *Cerastoderma glaucum* | 12/12 | $5.0 \pm 0.2$ | $-23.6 \pm 0.7$ | $-22.0 \pm 0.8$ | $5.7 \pm 0.4$ | $17.4 \pm 0.7$ |
| Polychaetes | 9/4 | $4.3 \pm 0.2$ | $-23.1 \pm 0.4$ | $-22.2 \pm 0.5$ | $11.0 \pm 0.6$ | $10.7 \pm 1.0$ |

The food sources were defined when a significantly different isotopic composition of at least one isotope existed. The differences of SI ratios among species were compared using a multivariate analysis of variance (MANOVA). Tukey's Honestly Significant Difference (HSD) test was used to detect significantly different means. Levene's test was used to test the homogeneity of variances.

## Selection of trophic enrichment factors

Different sets of trophic enrichment factors (TEFs) for carbon and nitrogen were used in a number of SI models (Table 2). For Model0, the carbon TEF was calculated for each food source individually by applying a function of $-0.199 \times \delta^{13}$C$_{source}-3.986$ as suggested by *Caut, Angulo & Courchamp (2009)*; the values ranged from $-0.2$ to 0.4‰ for individual species and/or combined sources. The standard error for the carbon TEF of the combined sources was determined by first-order error propagation of uncertainties (Appendix S1). The nitrogen TEF for bird blood was set at $2.25 \pm 0.20$‰ following *Caut, Angulo & Courchamp (2009)* who suggested the method to adjust isotope discrimination values for different consumer groups and their tissues according to the isotope composition of diet sources. As this method was criticized by *Perga & Grey (2010)* due to an inapplicable use of a variable TEF without specific knowledge of the predator–prey fractionation dynamics, we applied more sets of TEFs (Table 2) to assess sensitivity of inferences to variation in TEFs. In ModelA and ModelB, TEFs of carbon and nitrogen were used in order to prove the selection of the TEF values for Model0. In ModelC, we applied the TEF values obtained from *Caut, Angulo & Courchamp (2009)*, but used averaged single values (Table 2).

The mean reported trophic shift for sulphur ($0.5 \pm 0.56$‰) is not significantly different from zero (*Peterson & Fry, 1987*; *McCutchan et al., 2003*). Thus, we did not apply any TEF for sulphur in any of the SI models of this study.

## A Monte Carlo simulation of mixing polygons

A Monte Carlo simulation of mixing polygons (*Smith et al., 2013*) was used to apply the point-in-polygon assumption to the models. Convex hulls (*mixing polygons*) were iterated using distributions of dietary sources (Fig. 2) and different sets of TEFs (Table 2), and probabilities for consumers being in the mixing polygons were calculated. This provided a quantitative basis for consumer exclusion (those outside the 95% mixing region) or model rejection/validation.

**Table 2** Information about the models and applied simulations in the study.

| Model sets | Applied trophic enrichment factors | | | Validation by Monte Carlo simulations | Mixing model results | Prior information for mixing models |
|---|---|---|---|---|---|---|
| | $\delta^{13}C$ | $\delta^{15}N$ | $\delta^{34}S$ | | | |
| Model0 | −0.2 to 0.4‰ by[a] (see Appendix S1) | 2.25 ± 0.01‰[a] | Not applied (used as 0 ± 0 ‰) | Yes, but one individual was removed from further analysis | Yes | No WW[e] AFDW[f] |
| ModelA | 1.0 ± 0.2‰[b] | 4.5 ± 0.2‰[b] | | No; too many individuals lied outside the 95% mixing region or on its limit, and that requires alternative models to explain their isotopic signatures. | No | No |
| ModelB | 0.4 ± 0.17‰[c] | 2.67 ± 0.7‰[d] | | | No | No |
| ModelC | 0.17 ± 0.01‰ (as a mean of calculated TEFs based[a] | 2.25 ± 0.01‰ [a] | | Yes; one individual was removed from further analysis (the same one as in Model0) | Yes | No WW[e] AFDW[f] |

**Notes.**
[a] Using formula for C values and stated values for N (*Caut, Angulo & Courchamp, 2009*).
[b] *Federer et al., 2010* as an average between cellular blood and plasma.
[c] *McCutchan et al., 2003*.
[d] *Hobson, 2009*.
[e] Wet weight (WW) of different food objects from gut contents analysis.
[f] Organic matter weight (ash free dry weight; AFDW) of different food objects from gut contents analysis.

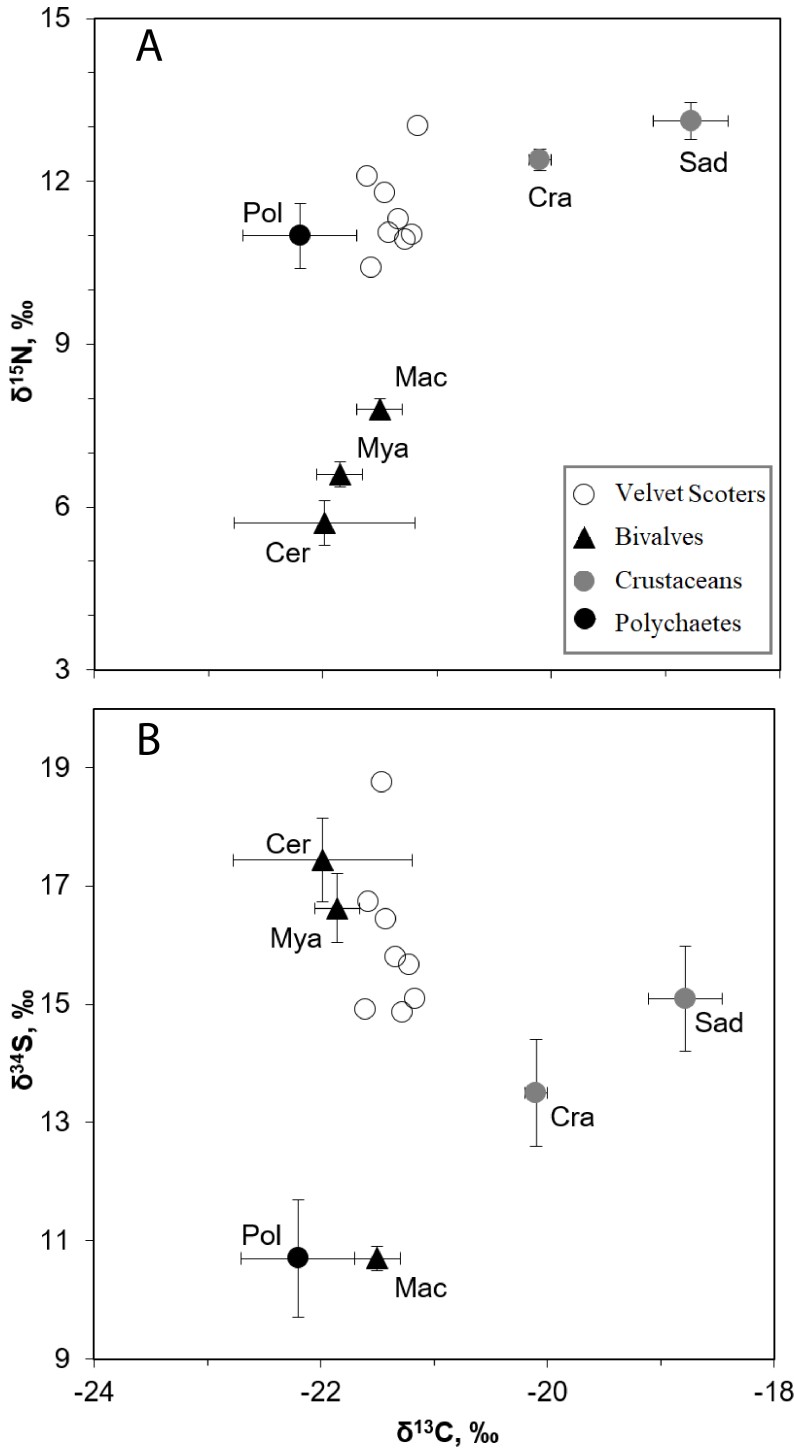

**Figure 2** **The mean $\delta^{34}$S, $\delta^{15}$N, and $\delta^{13}$C values (±SD) in the Velvet Scoters and potential food sources.** Open circles denote the Velvet Scoters. Triangles denote bivalves: Mac, *Macoma balthica*, Mya, *Mya arenaria*; Cer, *Cerastoderma glaucum*. Grey circles mark crustaceans: Sad, *Saduria entomon*; Cra, *Crangon crangon*. Black circle denotes polychaetes.

In Model0, one individual Velvet Scoter was excluded from further analysis (Figs. 3A–3F). That individual had higher $\delta^{34}$S values which were outside the 95% mixing region of the food sources. Consequently, Bayesian mixing models were calculated only for the seven Velvet Scoters that were determined to be within the 95% mixing region of the sources by three isotopes considered. As the TEF for carbon in ModelC differed only slightly from the one in Model0, the fit of both models to the mixing polygons were very similar. Thus, further Bayesian mixing modelling for ModelC were used for the seven Velvet Scoters (see mixing polygons, Appendix S3).

In ModelA and ModelB, relatively high TEFs affected the extents of mixing polygons which did not validate these models. Most consumers were characterized with very low probabilities to occur within the mixing polygons (Apendix S2 and S3). Thus, we rejected these models as unsuitable for diet estimation for the Velvet Scoters with the current food sources known to be available within the Lithuanian coastal zone.

## Stable isotope mixing models

Models, which were validated by Monte Carlo simulations of mixing polygons (i.e., Model0 and ModelC), were used for mixing modelling in the package SIAR (Stable Isotope Analysis in R; *Parnell et al., 2010*). The triple $\delta^{34}$S & $\delta^{15}$N & $\delta^{13}$C values were applied to estimate multiple food source contributions to composite diets. Additionally, we used three different information sets for mixing models: (A) no prior data, (B) prey proportions based on ash-free dry weight and (C) those based on wet weight as prior data from gut content analysis. The mean percentage with standard deviation (SD) and the 95% credibility interval ($CI_{95}$) were outputs from isotopic mixing models.

## RESULTS

### Diet composition by gut content analysis

Ninety-four percent of bycaught individuals contained at least some food remains in their esophagi and gizzards. Five species of soft bottom molluscs, two species of crustaceans, and benthic fish species were identified in the guts (Table 3). Soft bottom molluscs dominated in the diet, according to wet weights. *C. glaucum* bivalves dominated among the identified molluscs by wet weight, while the estimation of AFDW revealed that all three bivalve species were equally important in the diet. *S. entomon* were identified as important prey objects by estimations of both wet weight and AFDW. Fish only accounted for a trace portion of the prey items found in the gut content (Table 3). *C. glaucum* was the most frequent item, while half of the ducks also had other bivalves in their guts. *S. entomon* was consumed by the one third of ducks analysed.

### Stable isotope ratios of Velvet Scoters and their food sources

The SI ratios found within the blood samples of the eight Velvet Scoter individuals ranged by 0.4, 1.7 and 3.9‰ for $\delta^{13}$C, $\delta^{15}$N, and $\delta^{34}$S (Table 4). There were six main taxa of sandy bottom macrozoobenthos that significantly differed in isotopic composition (MANOVA, $F_{15,86} = 107.6$, $p < 0.05$; Fig. 2; Table 1). Because of similar values of three SI ratios bivalves, *C. glaucum,* and *M. arenaria* were pooled into one homogeneous group (HSD, $p > 0.05$).

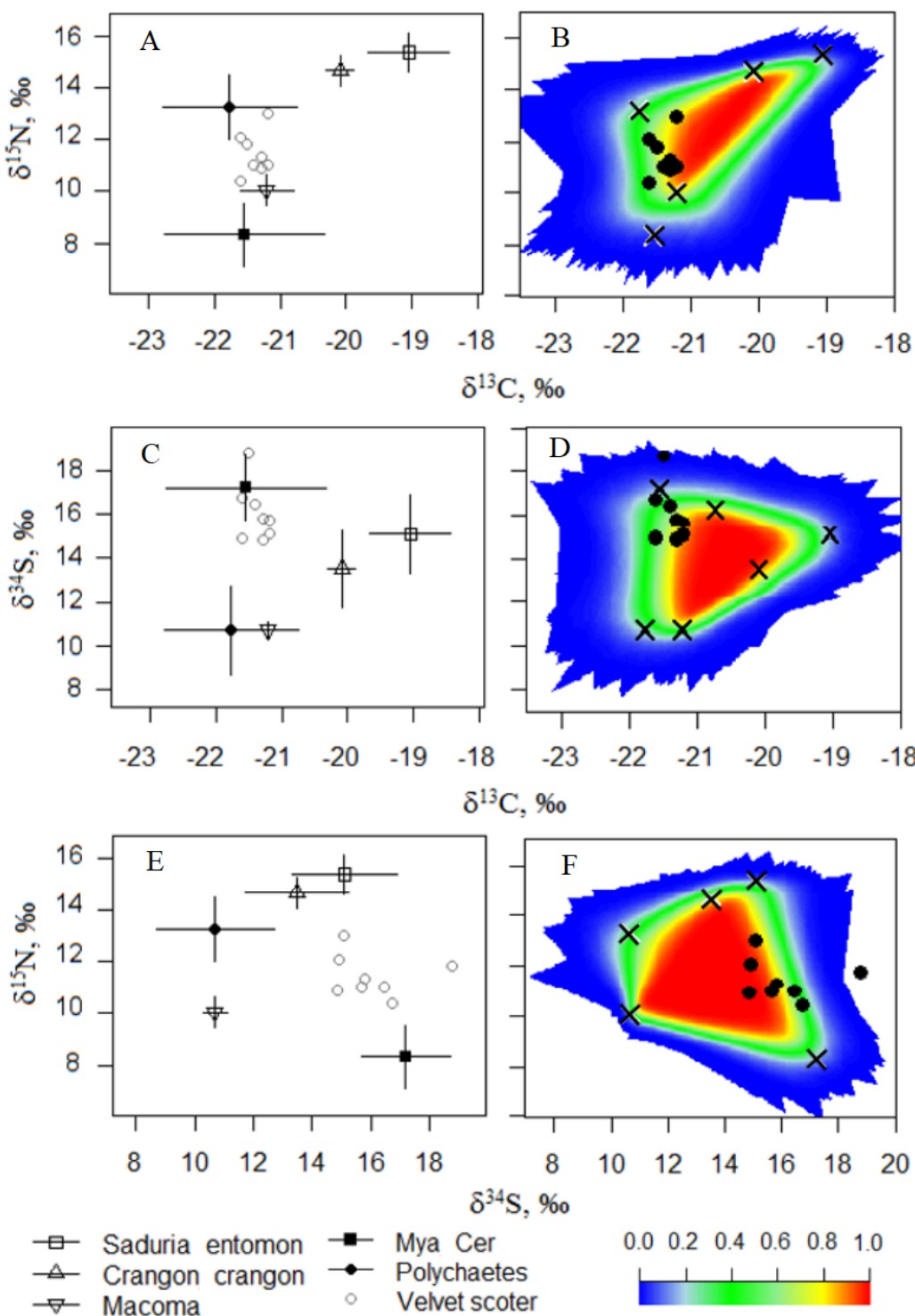

**Figure 3** (A, C, E) The five-source mixing model biplots with $\delta^{34}S$, $\delta^{15}N$, and $\delta^{13}C$ values after the TEF corrections in potential food sources and the Velvet Scoters (Model0). (B, D, F) The simulated mixing polygons for the biplots.

**Table 3 Diet composition of the Velvet Scoters (N = 66).** Wet weight (WW) and organic matter weight (AFDW) of different food objects in grams (g) and percent. Frequency of occurrence (FO) of prey objects by number of individuals (ind.) and percent of duck specimens which consumed particular prey.

| Taxa of prey objects | WW, g | WW, % | AFDW, g | AFDW % | FO, n | FO, % |
|---|---|---|---|---|---|---|
| Mollusca | | | | | | |
| *Mya arenaria* | 57.48 | 8.90 | 8.31 | 9.48 | 33 | 50.00 |
| *Macoma balthica* | 59.01 | 9.14 | 7.56 | 8.62 | 32 | 48.48 |
| *Cerastoderma glaucum* | 117.56 | 18.20 | 9.71 | 11.08 | 61 | 92.42 |
| *Rangia cuneata* | <0.01 | <0.01 | <0.01 | <0.01 | 1 | 1.52 |
| *Hydrobia ulvae* | <0.01 | <0.01 | <0.01 | <0.01 | 1 | 1.52 |
| Unident. Mollusca | 232.90 | 36.06 | 25.46 | 29.04 | 31 | 46.97 |
| Crustacea | | | | | | |
| *Crangon crangon* | 0.79 | 0.12 | 0.21 | 0.24 | 2 | 3.03 |
| *Saduria entomon* | 175.82 | 27.22 | 34.22 | 39.04 | 23 | 34.85 |
| Pisces | | | | | 2 | |
| *Ammodytes tobianus* | 2.25 | 0.35 | 2.19 | 2.50 | 2 | 3.03 |

**Table 4 Characteristics of analysed velvet scoters and their blood samples.**

| Number of analysed individuals | Body weight, g | C:N mass ratio | $\delta^{13}C$, ‰ | | $\delta^{15}N$, ‰ | | $\delta^{34}S$, ‰ | |
|---|---|---|---|---|---|---|---|---|
| | | | Min-Max | Mean | Min-Max | Mean | Min-Max | Mean |
| 8 | 1574 ± 128 | 3.5 ± 0.04 | −21.6–(−21.2) | −21.4 ± 0.2 | 10.4–13.0 | 11.5 ± 0.8 | 14.9–18.8 | 16.3 ± 1.3 |

The polychaetes and *M. balthica* had similar $\delta^{34}S$ values (HSD, $p > 0.05$), but might still be separated by $\delta^{15}N$ values (HSD, $p < 0.001$). The *C. crangon* and *S. entomon* crustaceans differed significantly in their $\delta^{13}C$ values (HSD, $p < 0.05$).

According to the defined SI values for the homogeneous groups, five benthic food sources could be distinguished: (1) *S. entomon*, (2) *C. crangon*, (3) *M. balthica*, (4) *M. arenaria* and *C. glaucum*, (5) polychaetes. These groups could be included as separate end-points into the mixing model.

## Mixing model results

The mixing models were run for Model0 (further description in the text and Table 5) and ModelC (Appendix S5; not described due to similarities to Table 5). They revealed that the main food sources for Velvet Scoters derived from the *M. arenaria* and *C. glaucum* group of bivalves, which contributed to 46 to 52% of the diet (Table 5; Figs. 4 and 5). The proportions of other food sources varied due to the different application of prior information into the mixing models. The prior information enhanced the importance of the *S. entomon* and *M. balthica*, and decreased the proportions of the *C. crangon* and polychaetes in diet estimations. Moreover, according to standard deviations and CI$_{95}$, prior information resulted in slightly more accurate diet estimates.

By comparing models results based on different prior information (Table 5), it was clear that inferences based on AFDW reduced the importance of the *S. entomon* to the diet of Velvet Scoters.

**Table 5  Contributions of food sources to the diet of the Velvet Scoters, which were calculated by the five-source mixing Model0.** Different sets of prior information as the wet weight (WW) or the organic matter weight (AFDW) of food objects from gut contents analysis were used for the mixing models.

| Sources | Proportions, % as Mean ± SD ($CI_{95}$) | | |
|---|---|---|---|
| | No prior information | WW | AFDW |
| *Saduria entomon* | 9 ± 7 (0–21) | 35 ± 4 (28–43) | 26 ± 5 (17–35) |
| *Crangon crangon* | 13 ± 9 (0–30) | 0.3 ± 0,5 (0–2) | 1 ± 1 (0–4) |
| *Mya arenaria & Cerastoderma glaucum* | 52 ± 9 (32–68) | 46 ± 4 (37–54) | 54 ± 5 (45–64) |
| *Macoma balthica* | 7 ± 6 (0–21) | 16 ± 3 (10–22) | 16 ± 3 (10–23) |
| Polychaetes | 18 ± 10 (0–36) | 3 ± 2 (0–6) | 2 ± 2 (0–5) |

## DISCUSSION

### Approaches of triple stable isotope measurements and gut content analysis for winter diet estimation for the Velvet Scoter

In this study, triple SI measurements and gut content analysis provided relevant estimates of the Velvet Scoter's diet in the wintering grounds of the Lithuanian Baltic Sea coastal zone. However, as the applied methods have specific limitations and require some assumptions, diet estimations might differ. Velvet Scoters, as other marine ducks, are mobile consumers and even in winter, when their foraging is largely restricted to the marine environment, they can move large distances as hydrological conditions change (*Cherel et al., 2008*). As the isotopic signature of tissues in newly arrived individuals might be acquired in previous feeding habitats (*Phillips & Gregg, 2001*), the SIA results should be interpreted with the assumption that the tissues analysed have reached an isotopic equilibrium before sampling at any particular wintering site. The isotopic half-life of the bird blood was estimated as being approximately two weeks, while complete equilibrium could take longer (*Vander Zanden et al., 2015*). Therefore, in this study, we checked the isotopic equilibrium in the blood of Velvet Scoters, according to SI ratios in food sources and different sets of TEFs (Fig. 3; according to *Smith et al., 2013*).

The selection of the most suitable TEFs for this particular study was a very important conjecture. It is known that TEFs may vary depending on a consumer's type, its nutritional status, diet quality, size, age, dietary ontogeny, tissue, elemental composition, and the isotopic value of their diet objects (e.g., *McCutchan et al., 2003*). We used a method by *Caut, Angulo & Courchamp (2009)* to calculate TEFs for carbon and nitrogen from the SI ratios of food sources, depending on the consumer classes and types of tissue. As this method was found to be contradictory (*Perga & Grey, 2010*), we also showed the effects of different sets of TEFs to final estimations about the winter diet for Velvet Scoters. Model0 assumed TEFs suggested by *Caut, Angulo & Courchamp (2009)* (i.e., varied TEFs for carbon according SI values of the selected food sources; Table 2). Model A and B assumed higher TEFs than Model0, but they were relevant for marine ducks (*McCutchan et al., 2003*; *Hobson, 2009*; *Federer et al., 2010*). ModelC was run with mean TEFs for carbon, as also suggested by *Caut, Angulo & Courchamp (2009)*. According to Monte Carlo simulations for a priori evaluation of mixing models, we omitted ModelA and ModelB as unsuitable for estimation of Velvet Scoter winter diets (*Smith et al., 2013*). In the cases of Model0

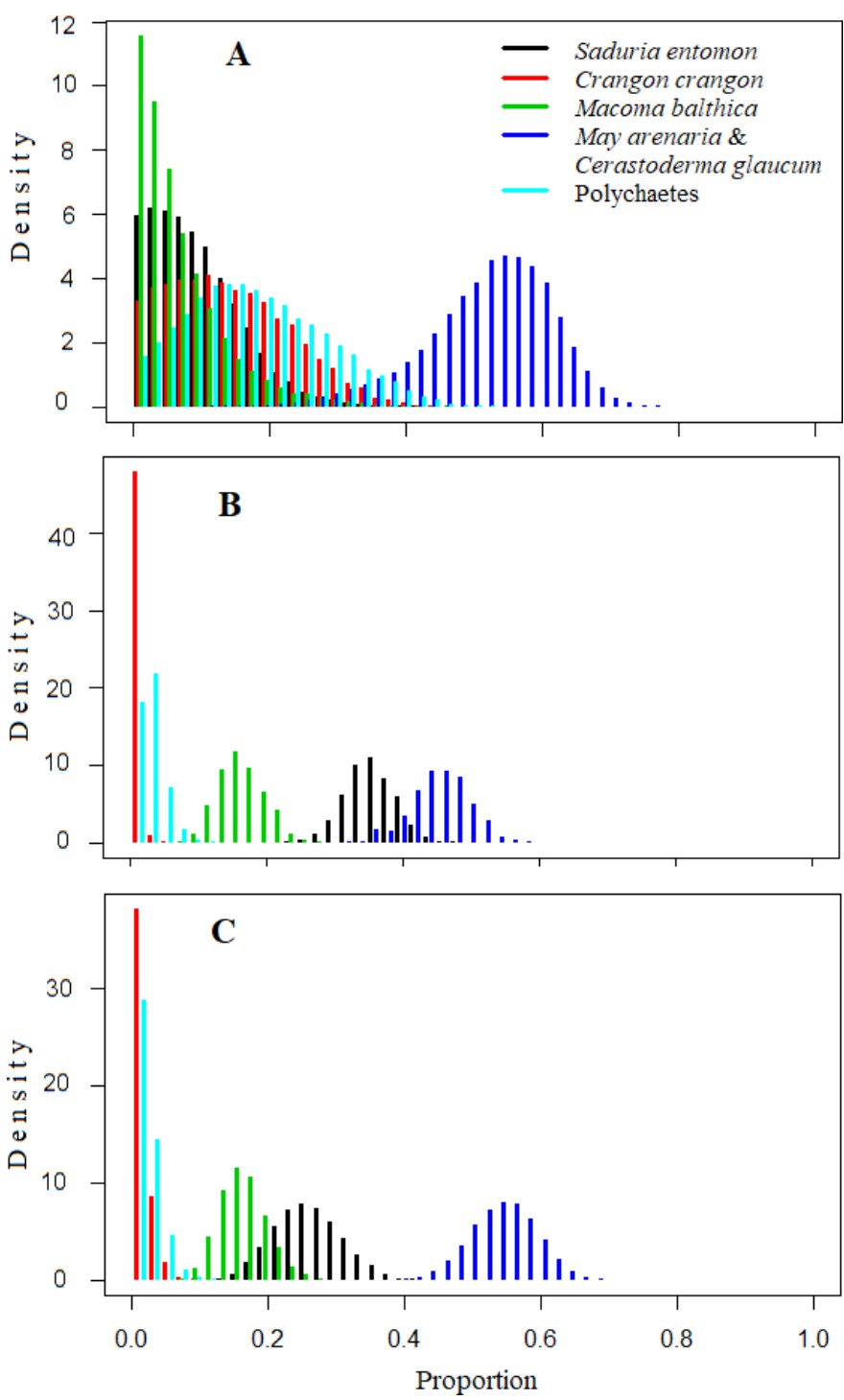

**Figure 4 Density histograms showing estimated contribution of food sources for seven velvet scoters (Model0).** (A) The model without prior information on diet. (B) The model with organic matter weight (ash free dry weight; AFDW) and (C) the model with wet weight (WW) of different food objects from gut contents analysis as prior information.

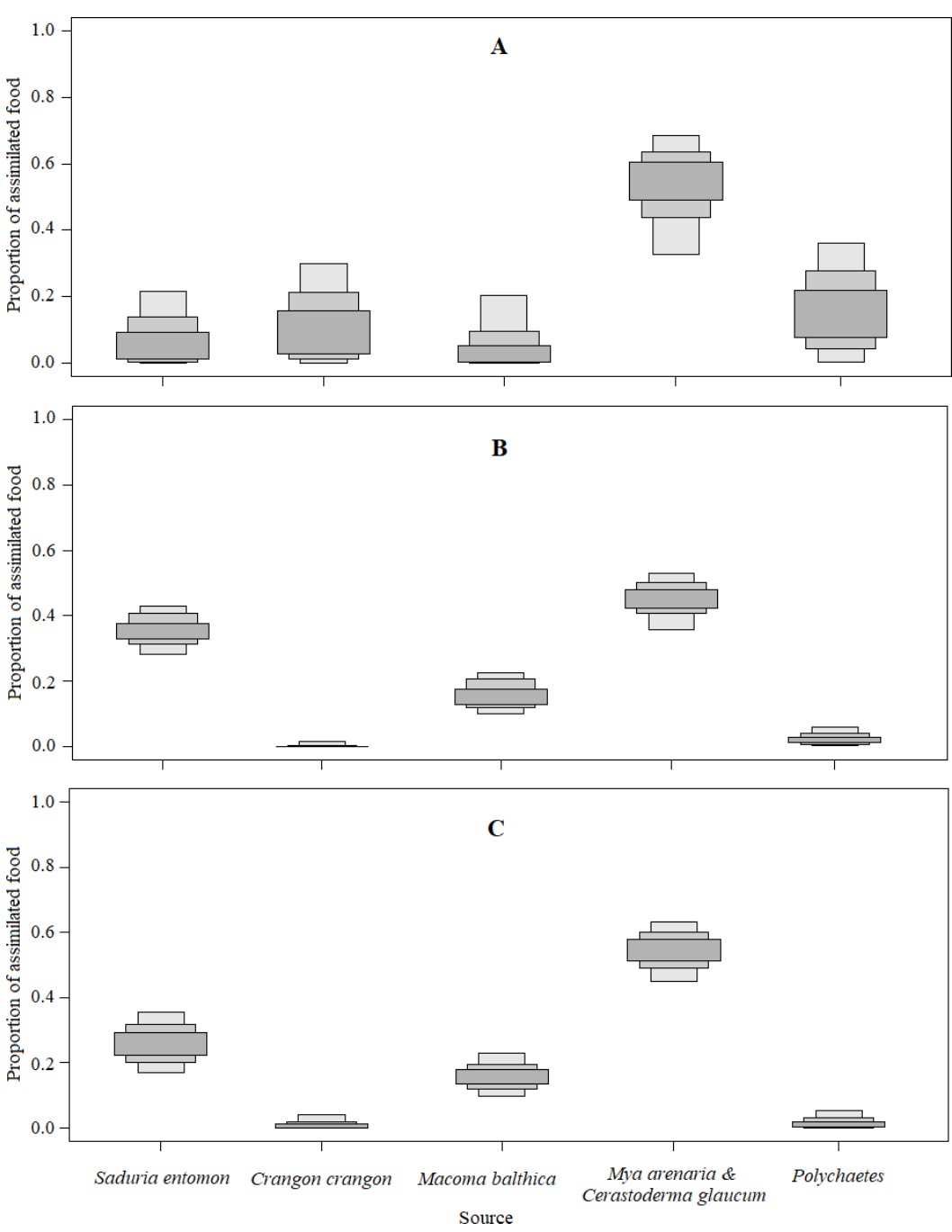

**Figure 5 The estimated relative contributions of food sources (Model0).** Each plot shows 50% (dark grey), 75% (medium grey), and 95% (light grey) Bayesian credibility intervals of contributions of each source. (A) The model without prior information on diet. (B) The model with organic matter weight (ash free dry weight; AFDW) and (C) the model with wet weight (WW) of different food objects from gut contents analysis as prior information.

and Model C, one of eight ducks was eliminated from further diet analysis due to possible non-equilibrium of $\delta^{34}$S ratios to local food sources in the Lithuanian coastal ecosystem. We also applied external information about gut content compositions which were collected during this study using ducks that had been caught by fishermen. As their gut contents were assessed according to the proportions of the wet weights and AFDW of the prey items, we used both these estimations as prior information for the SI mixing models. Moreover, even though some potential prey items (e.g., polychaetes) were not detected during gut content analysis, their high frequency of occurrence in guts of marine ducks had been documented previously (*Žydelis, 2002*), so we still considered them as source material in our SI mixing models for the evaluation of the Velvet Scoter diet within the Lithuanian coastal zone.

Estimates of food source proportions were very similar between Model0 and ModelC, from which the maximum difference for the proportions of the food sources was 2% (see Table 5 and Appendix S5). The difference between the mixing model results was negligible due to the relatively low variability of carbon TEFs found among the different food sources. Thus, we conclude that even if we apply the varying TEF for carbon (according to *Caut, Angulo & Courchamp, 2009*), this variation was sufficiently low (from $-0.2$ to $0.4\permil$ ), that it did not affect mixing model results.

As SIA is based on previously known and potential diet estimations, gut content analysis is assumed to be crucial for the taxonomic identification of prey objects. In this study, one single individual of the invasive *Rangia cuneata* bivalve species was found within the guts of a Velvet Scoter. This bivalve was first reported in Lithuanian waters in 2013 (*Solovjova, 2017a*; *Solovjova, 2017b*), and the current study has confirmed that *R. cuneata* plays a role in the coastal food web.

## Application of $\delta^{34}$S ratios

This study showed that analysis of the $\delta^{34}$S ratios increased the capacity to discriminate a higher number of macrozoobenthos taxa for modelling the food source contributions in the diet of benthivorous Velvet Scoter. Benthic invertebrates obtain their sulphur from either sediments, the below sediment-water interface, or the water column, and this could be the reason for taxa-specific $\delta^{34}$S values (*Croisetiere et al., 2009*; *Karube, Okada & Tayasu, 2012*). Unfortunately, the homogenous SI values found in *M. arenaria* and *C. glaucum* did not allow for further discrimination, and therefore, they were aggregated for further use in the SI mixing model. However, in using $\delta^{34}$S values, we could distinguish polychaetes and *M. balthica* from the other bivalves and crustaceans, which might be explained by their different use of organic material. *M. balthica* might be attributed to switches between suspension- and deposit-feeding (*Zwarts & Wanink, 1989*; *Lin & Hines, 1994*) and this might be reflected in their sulphur isotopic composition. We have found that facultative suspension feeders, such as *M. balthica* and polychaetes, had approximately $5.5\permil$ lower $\delta^{34}$S values than the obligatory suspension feeders such as *C. glaucum* and *M. arenaria*. Moreover, in this study, polychaetes had much higher $\delta^{15}$N values than *M. balthica* (the difference was $3.5\permil$ ), which reflected their higher trophic position in the food web relative to the primary sources of organic matter available. Therefore, the triple isotope approach

allowed the relatively precise discrimination of the main macrozoobenthos organisms as food sources for the Velvet Scoter.

## Estimation of the winter diet of the Velvet Scoter

The results of the winter diet composition of Velvet Scoters, which were estimated using both the triple SI approach and the gut content analysis, were comparable and complementary. Both methods revealed the preference by Velvet Scoters for the *M. arenaria* and *C. glaucum*, while the proportions of other food sources varied. The joint contribution of *C. glaucum* and *M. arenaria* comprised approximately half of Velvet Scoter's diet (Table 5), while *M. balthica* was only responsible for 7 to 16% of their diet. This result differed from a previous study during 1996–2002, which showed the dominance of *M. arenaria* for 82% of the total wet weight content found in the gut of Velvet Scoter (*Žydelis, 2002*). Although previously, *C. glaucum* had not been reported as prey items for scoters in the Lithuanian coastal zone (*Žydelis, 2002*), it was consumed by 92% of total number of Velvet Scoters analysed in this study (Table 3). Moreover, *C. glaucum* has been reported as one of the dominant prey items in their diet along the Danish, English, Polish, and German Baltic coasts (a review by *Fox, 2003*).

As the number of certain prey species might vary temporally, the diet composition of Velvet Scoters reflects this variability (*Fox, 2003*). The biomass of *C. glaucum* increased from $0\,\mathrm{g\,m^{-2}}$ in 1996–2002 to more than $18\,\mathrm{g\,m^{-2}}$ in 2012–2016 (while more than $100\,\mathrm{g\,m^{-2}}$ in 2014) within a depth range of 13–15 m at Juodkrantė, Lithuania (State monitoring data of the Marine Research Department under the Environmental Protection Agency; *Solovjova, 2017a*; *Solovjova, 2017b*). Therefore, the differences in the diet compositions of the Velvet Scoter estimated by *Žydelis (2002)* and this study could be explained by possible shifts in the biomasses and proportions of the prey species available to Velvet Scoters between the two periods.

The results of our SI mixing model revealed that the *S. entomon* contributed 9% towards the diet of the Velvet Scoter, while the gut content analysis revealed a contribution of 36% by wet weight and 29% by AFDW. Using data from gut content analyses as prior information, SI mixing models revealed the higher importance of this crustacean to the Velvet Scoter diet (by 35% by wet weight and 26% by AFDW; Table 5). Previous gut content analyses only showed a small contribution of *S. entomon* to the Velvet Scoter's diet (3% of total wet weight; *Žydelis, 2002*), but it was an important prey item for the Long-tailed duck over the same sandy bottom habitat (74% of total wet weight) (*Žydelis & Ruškyte, 2005*). *S. entomon* is abundant in deeper areas compared to the inshore coastal zone, so ducks that feed on this prey might do so in deeper waters; this may be especially so in the northern part of Lithuanian marine waters, where a marine protected area was established due to high and regular marine bird abundances, including Velvet Scoters (*Daunys et al., 2015*). Therefore, the number of *S. entomon* in the coastal zone and its importance to the feeding of marine ducks might differ during the course of winter when they come closer to the coast and among other years, depending on the hydrological conditions (e.g., *Bacevičius, 2013*).

As bird gut content analysis is based on the weights of objects found in the gizzard and esophagus, it is common to overestimate indigestible items or those that are more difficult

to digest which could contribute to the total weight of prey items. Conversely, soft-bodied prey as polychaetes are often underestimated because of their rapid digestion in the foregut and lower detection probability, which is further influenced by the proficiency of the researchers (review of *Žydelis & Richman, 2015*). In this study, we did not find polychaetes in any ducks examined, but the SI mixing models, without prior information, estimated their contribution of 18% to the diet. Their inferred importance declined considerably to 2% when using gut-based information in the process of mixing modelling (Table 5). Moreover, polychaetes have been mentioned as common foods for marine ducks by other authors; e.g., *Žydelis (2002)* reported that polychaetes were taken by 83% of all the Velvet Scoters studied, but contributed only 3% to the total wet weight.

The energy/caloric value of the prey is important determinant of their nutritional value for marine ducks in winter. Bivalves are of low caloric value with a high inorganic indigestible content (*Fox, 2003*). Moreover, crushing the hard shell of *C. glaucum* might require more energy in comparison to the lighter shell of *M. balthica* and *M. arenaria* (*Rumohr, Brey & Ankar, 1987*). Scarcer but more easily digestible prey items such as polychaetes or fish could provide a greater energy/caloric value than bivalves (review of *Žydelis & Richman, 2015*). This might account for the apparent differences in diet estimates provided by SI mixing models vs. gut content analysis in this study. Moreover, the SIA provides information on assimilated (not only ingested) food items and assumptions on the importance of other prey items, such as soft-bodied prey which are usually underestimated during gut content analysis. This is important because the food items of benthivorous ducks differ from each other by energy/caloric values and may have already undergone temporal physiological changes (*Waldeck & Larsson, 2013*).

This study was based on a relatively low sample size of live bird blood samples due to inherent difficulties of catching live birds in their marine wintering grounds in the open coastal zone. A larger sample of Velvet Scoters for SIA analysis of blood would likely improve precision of estimates and permit a comparison of diets inferred for different sex and age groups, as well as uncover potentially important temporal and spatial variation in winter diets.

## CONCLUSIONS

In this study, we demonstrated how information about diet composition can be obtained using non-lethal blood sampling from live ducks, gut content analysis of bycaught individuals, and triple SI mixing modelling. Moreover, we also illustrate the benefits of the application of the $\delta^{34}$S ratio as complementary to the $\delta^{13}$C and $\delta^{15}$N ratios in discriminating sandy bottom macrozoobenthos organisms with obligatory and facultative suspension feeding in the Baltic Sea.

The results revealed the main contribution of the group of *M. arenaria* and *C. glaucum* to be 46–54% of the Velvet Scoter's diet. The *S. entomon* contributed one third towards the diet, while other food sources accounted for the rest. We also discussed possible diet shifts by Velvet Scoters from changes in feeding habitats. Questions on methods to study the diet composition and its temporal changes should be taken into account when analysing the strong decline in the number of wintering marine ducks in the Baltic Sea.

## ACKNOWLEDGEMENTS

An important acknowledgement goes to coastal fishermen who shared bycaught carcasses for diet research. Authors thank the bird catchers from the EU LIFE+NATURE DENOFLIT project (LIFE09 NAT/LT/000234) for the possibility to take samples from live marine birds. Thanks to dr. Mindaugas Dagys, Gintaras Riauba, and Agnė Račkauskaitė for help with bycaught birds in laboratories. Information of benthos biomasses was collected by Sabina Solovjova from the Marine Research Department under the Environmental Protection Agency, Lithuania. Dr. Ramūnas Žydelis and Dr. Ray Alisauskas offered suggestions that improved the manuscript. The authors appreciate comments and useful recomendations by Dr. Keith A Hobson, other two anonymous reviewers, and the academic editor Dr. Perran Cook.

### Funding

This study was supported by Klaipėda University and the project Lithuanian Maritime Sector's Technologies and Environmental Research Development (No VP1-3.1-ŠMM-08-K-01-019). The funders had no role in study design, data collection and analysis, decision to publish, or preparation of the manuscript.

### Grant Disclosures

The following grant information was disclosed by the authors:
Klaipėda University.
Lithuanian Maritime Sector's Technologies and Environmental Research Development: VP1-3.1-ŠMM-08-K-01-019.

### Competing Interests

The authors declare there are no competing interests.

### Author Contributions

- Rasa Morkūnė conceived and designed the experiments, performed the experiments, analyzed the data, contributed reagents/materials/analysis tools, prepared figures and/or tables, approved the final draft.
- Jūratė Lesutienė conceived and designed the experiments, performed the experiments, contributed reagents/materials/analysis tools, authored or reviewed drafts of the paper.
- Julius Morkūnas conceived and designed the experiments, performed the experiments, analyzed the data, contributed reagents/materials/analysis tools, authored or reviewed drafts of the paper.
- Rūta Barisevičiūtė conceived and designed the experiments, performed the experiments, contributed reagents/materials/analysis tools.

### Field Study Permissions

The following information was supplied relating to field study approvals (i.e., approving body and any reference numbers):

Permits to capture, use, and release protected bird species for scientific research were obtained from the Environmental Protection Agency of Lithuania (permit 2012 No. 7 and 2013 No. 1).

## Data Availability

The raw data are provided in a Supplemental File.

## Supplemental Information

Supplemental information for this article can be found online at http://dx.doi.org/10.7717/peerj.5128#supplemental-information.

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
