# Peer review of "Triple stable isotope analysis to estimate the diet of the Velvet Scoter (Melanitta fusca) in the Baltic Sea"

_PeerJ, doi:10.7717/peerj.5128_

## Round 0.1 · original submission · Major Revisions

Thank you for submitting this manuscript to PeerJ. All three reviewers are supportive of publication however a number of key issues need to be addressed first.

The text needs to be edited by a fluent or native English speaker to improve the clarity and flow. Perhaps a professional service could be engaged if necessary?

Reviewers 2 and 3 both make some key points about how you chose your TEF values and suggest these be better justified. I note reviewer 2 makes some particularly helpful comments in this regard.

Reviewer 2 raises some good points about spatial variability and consistency of sample preparation that should be addressed.

Reviewer 1 ·

Basic reporting

The article is written in good, professional English but a further revision by a native speaker would be desirable.

Literature references and sufficient field background/context are provided.
A professional article structure is provided. Raw data are shared.

Figures are relevant to the content of the article, but figure and table legends are missing.

Self-contained with relevant results.

Experimental design

no comment

Validity of the findings

no comment

Additional comments

A useful and well designed study. Can you add a figure comparing the results of the two methods (e.g. pie chart)?

Reviewer 2 ·

Basic reporting

1. Grammatical errors are present throughout the manuscript and the manuscript would require close editing by a fluent English speaker prior to it being suitable for publication.

2. Please increase the amount of information included in Figure and Table legends. Both should have sufficient information to make the Figure or Table interpretable and understandable without reference back to the text. So all species names, abbreviations and units should be clearly described.

Experimental design

1. Spatial information is central to many isotope studies and is often the key to understanding why animals show the isotope values they do. I raise this point because the isopod crustaceans were sampled c. 40km from the ducks and other prey and were also quite a bit deeper (c. 35m vs 15-20m; Figure 1). Differences of 15-20m can potentially have a strong influence on C and N isotope values. Were there no isopods to be sampled near the ducks? Presumably not, but the authors need to be explicit about the fact that the spatial offset of the different prey is a potential source of variability in their results.

Validity of the findings

1. Consistency in sample preparation is key for most stable isotope applications. In this case, the blood of the ducks was freeze-dried but the invertebrates were dried in a conventional oven. There is literature that suggests freeze drying and oven drying can yield different N isotope values for the same organism/tissue (Carabel et al. 2006). Using different methods on predators vs prey could pose yet another potential source of variability that must be addressed, and if not corrected for, at the very least described and considered. It is incumbent on the authors to defend the use of different preservation techniques either by reanalyzing a subset of samples to verify there is no difference, going to the relevant literature and finding suitable mathematical conversions if they exist, or finding previous studies that support their current approach.

2. The use of the Caut et al. approach to estimating prey-specific TEF values is not widely accepted. Perga and Grey (2010) provide convincing counter arguments against the blind use of a variable TEF without specific knowledge of the predator-prey fractionation dynamics. I think most readers would question the application of the approach here, or at the very least, want to see what effect using the Caut et al. model had on the final outcome. Did it alter the number of scoter retained from the mixing model polygon analysis? To what extent did it change the estimated dietary proportions from the different prey groups using SIAR? I would suggest that the apparent dearth of information on velvet scoter TEFs would suggest a more conservative approach based on a single, common TEF appropriate for avian blood (such values are documented in numerous waterbird-specific studies as well as several meta-analyses on the subject). If the authors feel strongly that the Caut et al. variable TEF model is the most appropriate, then I would strongly suggest running the analysis again with a single generic TEF, and describing the differences (or better, providing the results as supplemental material) so that readers are aware of how this approach influenced the findings reported in the paper. I also think that there needs to be an acknowledgement that this approach is controversial and a statement given as to why the authors feel the approach is justified here.

Additional comments

I want to encourage the authors to pursue this work, particularly the approach of using three isotopes and multiple statistical 'checks and balances' to study the trophic ecology of this duck species. Coupling the isotope work with the diet analysis is powerful and a strong component of this study.

·

Basic reporting

This is an interesting paper and one of few that have explored the use of stable-sulfur isotope measurements in seaducks to infer dietary patterns. In general, the reporting is clear and straightforward but there are several cases where the English can be improved and the manuscript would benefit from editing by a native English speaker. Nonetheless, the writing is still perfectly intelligible and unambiguous.
The literature cited is generally relevant but the reliance on the Caut et al. approach to assigning TEF values could have benefitted from reporting of derived blood TEF values for species with similar ecology(Canvasback: Haramis et al. Auk 118:1008-1017;King Eiders: Opel and Powell J of Ornithol. 151:123-131; Spectacled Eiders: Federer et al. Cdn J, Zool 88:866-874; see also the paper by Lovorn et al on isotopes in eiders in the Chuckchi Sea, Progress in Oceaography 136:162-174).
The table and figures are appropriate although I did expect to see the Siar or MixSiar output probability distributions for each of the relevant dietary inputs.
There were no significant hypotheses stated as this was largely an exploratory study.

Experimental design

The paper presents the results of original research in keeping with the aims of the journal.
The research question is extremely straightforward (namely use a 3 isotope mixing model to estimate dietary inputs to scoters). Moreover, justification for the study was clearly the extreme population decline of the species in these waters.
The authors generally followed a well defined protocol of applying the SIAR model. This requires estimates of TEF, isotopically distinct dietary endpoints etc. I think the presentation could have been improved by applying variance estimates about the TEF values (if done, I did not detect that). That would have made better use of the Bayesian approach to propagating error. As stated above,the TEF estimates were adopted rather uncritically and the specific diving duck work done in captivity needs to be considered or at least presented to see how well they fit the Caut estimate (for the carbon and nitrogen isotopes, I agree that an assumption of no TEF for sulfur seems appropriate). On a similar note, the authors need to quote the credibility intervals for their predictions throughout (not just the means). Stable isotope methods reveal that largely inorganic standards were used in the lab and this is frowned upon when measuring organic materials (principle of like materials), although I doubt if this will influence the results. Would have been nice to ground truth the Post lipid correction algorithm by simply running a few samples with lipids actually removed. I have much less faith in the C:N correction approach.

Validity of the findings

The findings seem to confirm other, non-isotopic, data regarding scoter diet.
Unfortunately, the sample size for the live-caught birds is extremely low and it is just not clear how well this study represents the bulk of wintering birds. On that note, I was surprised to determine that the authors did not make use of the dead scoters from the bycatch. Presumably, their muscle and liver tissue would have been excellent for dietary reconstruction using stable isotopes. In addition to presenting results of the stomach content analyses, SIAR and, more currently MixSiar could readily use those data as a Bayesian prior in the diet estimate analyses.Not at all sure why the authors did not do that.

Overall, the paper could have ended with a stronger concluding statement regarding the importance of these findings. How exactly are such findings relevant to deciphering the declines of scoters in this area?

Additional comments

This is a nice study. I do recommend that you use the MixSiar package and provide the output that shows the probability distributions of various dietary inputs. It would also be instructive to see how the estimates change when using the Bayesian prior based on the gut contents analysis. Do bring the paper up-to-speed by including directly the isotope studies done to date on diving ducks (mostly in the North American literature).

---

## Round 0.2 · Minor Revisions

Thank you for responding to the reviewers comments. Reviewer 2 has indicated s/he believes you have addressed the comments to their satisfaction.

I have read over the text and it is greatly improved, thank you for doing this. I have added a few minor comments and suggested edits in the attached pdf for you to consider.

Reviewer 2 ·

Basic reporting

The authors have sufficiently addressed all concerns.

Experimental design

The authors have sufficiently addressed all concerns.

Validity of the findings

The authors have sufficiently addressed all concerns.

Additional comments

Thank you for your obvious efforts in responding to the reviewers' comments.

---

## Round 0.3 · Minor Revisions

Thank you for making the changes I suggested.

I have consulted with the Section Editors and the journal management and they believe that although the article is now scientifically acceptable, the language is still unclear in places and could be improved further. Please could you carefully edit the manuscript, ideally with the assistance of a fluent English speaker to further improve the clarity.

---

## Author Rebuttal · Round 0.3

Marine Research Institute
Klaipėda University
Herkaus Manto str. 84,
92294 Klaipeda
Lithuania

9 May, 2018

Dear Editor,

Thank you for your suggestions to improve the manuscript. We have edited the text and hope that the manuscript is now suitable for publication in PeerJ.

Your Sincerely,
Dr. Rasa Morkūnė
Klaipėda University

---

## Round 0.4 · Minor Revisions

Thanks for having a native speaker read over the manuscript, this has significantly improved the language. I have been over it one more time and suggest a few more final edits in the annotated pdf attached.

---

## Round 0.5 · accepted · Accept

Thanks for improving the text. The article is now acceptable for publication.

#